# The acceptability of the AMBITION-cm treatment regimen for HIV-associated cryptococcal meningitis: Findings from a qualitative methods study of participants and researchers in Botswana and Uganda

**David S. Lawrence**[1,2]*, **Agnes Ssali**[3,4], **Neo Moshashane**[2], **Georgina Nabaggala**[3], **Lebogang Maphane**[2], **Thomas S. Harrison**[5,6,7], **David B. Meya**[8], **Joseph N. Jarvis**[1,2], **Janet Seeley**[3,4]

**1** Department of Clinical Research, Faculty of Infectious and Tropical Diseases, London School of Hygiene and Tropical Medicine, London, United Kingdom, **2** Botswana Harvard AIDS Institute Partnership, Gaborone, Botswana, **3** Social Aspects of Health Programme, MRC/UVRI & LSHTM Uganda Research Institute, Entebbe, Uganda, **4** Department of Global Health and Development, Faculty of Public Health and Policy, London School of Hygiene and Tropical Medicine, London, United Kingdom, **5** Institute of Infection and Immunity, St George's University London, London, United Kingdom, **6** Clinical Academic Group in Infection and Immunity, St George's University Hospitals NHS Foundation Trust, London, United Kingdom, **7** MRC Centre for Medical Mycology, University of Exeter, Exeter, United Kingdom, **8** Infectious Diseases Institute, Makerere University, Kampala, Uganda

* david.s.lawrence@lshtm.ac.uk

## Abstract

### Background

The AMBITION-cm trial for HIV-associated cryptococcal meningitis demonstrated that a single, high-dose of liposomal amphotericin (AmBisome) plus 14-days of oral flucytosine and fluconazole was non-inferior in terms of all-cause mortality to 7-days of amphotericin B deoxycholate and flucytosine followed by 7-days of fluconazole (Control). The AmBisome regimen was associated with fewer adverse events. We explored the acceptability of the AmBisome regimen from the perspective of participants and providers.

### Methods

We embedded a qualitative methods study within the AMBITION-cm sites in Botswana and Uganda. We conducted in-depth interviews with trial participants, surrogate decision makers, and researchers and combined these with direct observations. Interviews were transcribed, translated, and analysed thematically.

### Results

We interviewed 38 trial participants, 20 surrogate decision makers, and 31 researchers. Participant understanding of the trial was limited; however, there was a preference for the AmBisome regimen due to the single intravenous dose and fewer side effects. More time was required to prepare the single AmBisome dose but this was felt to be acceptable given

**Data Availability Statement:** Descriptive data of the participants and relevant qualitative data related to the findings are presented in the manuscript. Material related to the study are accessible at LSHTM Data Compass at https://datacompass. lshtm.ac.uk/id/eprint/3106/. Requests to access additional anonymised data will be evaluated on a case-by-case basis upon request to researchdatamanagement@lshtm.ac.uk.

**Funding:** This study was funded by the National Institute for Health Research (NIHR) through a Global Health Research Professorship to JNJ (RP-2017-08-ST2-012) using UK aid from the UK Government to support global health research, including salary support for NM, GN and JNJ. The AMBITION-cm trial, including salary support for DSL, DM, TSH and JNJ was supported by a grant through the European Developing Countries Clinical Trials Partnership (EDCTP), the Swedish International Development Cooperation Agency (SIDA) (TRIA2015-1092), and the Wellcome Trust / Medical Research Council (UK) / UKAID Joint Global Health Trials (MR/P006922/1). The views expressed in this publication are those of the authors and not necessarily those of the funders and the funders had no role in study design, data collection and analysis, decision to publish, or preparation of the manuscript.

**Competing interests:** The authors have declared that no competing interests exist.

subsequent reductions in workload. The AmBisome regimen was reported to be associated with fewer episodes of rigors and thrombophlebitis and a reduction in the number of intravenous cannulae required. Less intensive monitoring and management was required for participants in the AmBisome arm.

## Conclusions

The AmBisome regimen was highly acceptable, being simpler to administer despite the initial time investment required. The regimen was well tolerated and associated with less toxicity and resultant management. Widespread implementation would reduce the clinical workload of healthcare workers caring for patients with HIV-associated cryptococcal meningitis.

## Author summary

The AMBIsome Therapy Induction OptimisatioN (AMBITION-cm) clinical trial found that a single, high-dose, intravenous liposomal amphotericin B (AmBisome) based regimen for HIV-associated cryptococcal meningitis was non-inferior to the WHO recommended first-line treatment which includes seven daily doses of intravenous amphotericin B deoxycholate. The AmBisome regimen was also associated with fewer adverse events. In addition to the clinical efficacy data it is important to consider how acceptable the AmBisome regimen was from the perspectives of those who received the regimen as well as the healthcare workers administering it. To do this we conducted a qualitative methods study of in-depth interviews with AMBITION-cm trial participants, surrogate decision makers, and researchers working on the trial. These interviews were combined with direct observations of the research process and analysed thematically. The trial participants were often severely unwell and therefore the understanding of the trial was limited; however, the AmBisome regimen was generally preferred due to the single intravenous dose and fewer side effects. Researchers strongly preferred the AmBisome regimen which took less time to administer overall and was also associated with fewer side effects. We conclude that these findings complement the efficacy data from the clinical trial to support widespread implementation of the regimen.

## Background

HIV-associated cryptococcal meningitis remains a significant driver of AIDS-related mortality. There are an estimated 152,000 cases of cryptococcal meningitis each year, the majority of which occur in sub-Saharan Africa [1]. Cryptococcal meningitis is estimated to be responsible for 112,000 deaths annually and is the cause of 19% of all AIDS-related deaths. The burden of cryptococcal meningitis persists despite widened access to antiretroviral therapy (ART), with recent programmatic data from South Africa and Botswana indicating that the number of cases has stayed relatively constant in recent years [2,3]. Cryptococcal meningitis primarily affects people with advanced HIV disease, typically with a CD4 count less than 100 cells/uL, and there remains a relatively constant population of people living with HIV who are diagnosed with advanced disease either as a result of delayed diagnosis or treatment failure due to difficulties with adherence and/or drug resistance [4,5].

Outcomes among patients diagnosed with cryptococcal meningitis are often poor. This is due to factors including presenting to care with severe disease, inadequate antifungal therapy,

and drug-related toxicities. Cryptococcal meningitis has historically been treated with high-dose oral fluconazole monotherapy which is widely available but associated with high mortality: over 50% at ten weeks and over 70% within a year [6–9]. Ten week mortality outcomes can be improved to roughly 40% in clinical trial settings when combining fluconazole with 14 daily doses of intravenous amphotericin B deoxycholate (amphotericin B) [10,11] but this regimen is notoriously toxic and prolonged courses often lead to renal impairment, electrolyte disturbances, anaemia and thrombophlebitis [12,13]. An alternative antifungal, flucytosine, which is given for seven to fourteen days in four daily oral doses has been proven to be superior to fluconazole as a partner drug for amphotericin B [10]. The enhanced antifungal effect of flucytosine permits a reduction in the duration of amphotericin B from 14 to seven days, mitigating but not eliminating amphotericin-related toxicities [10]. These toxicities can be further reduced when managing patients with intravenous fluid administration both before and after each daily amphotericin B infusion, and oral potassium and magnesium supplementation, but they cannot not be eliminated. The administration of seven days of amphotericin B and the pre-emptive medication, as well as the monitoring and management of drug-related toxicity, remains complex and require intensive time and resources from healthcare professionals, as well as contributing to poor outcomes among patients.

Liposomal amphotericin (AmBisome, Gilead Sciences Inc) is associated with fewer drug-related toxicities [14–16] and has been proven to be well suited to single, high-dose administration in both cryptococcal meningitis [17] and other infections [18,19]. The AMBIsome Therapy Induction OptimisatioN (AMBITION-cm) trial was a non-inferiority phase-III trial of a single, high-dose of AmBisome given with 14 days of flucytosine and fluconazole in comparison to the World Health Organisation defined standard of care: 7 days of amphotericin B given with 7 days of flucytosine and followed by 7 days of fluconazole [20]. AMBITION-cm recruited 844 participants from eight hospitals in five countries: Botswana, Malawi, South Africa, Uganda, and Zimbabwe. A total of 814 participants were included in the intention-to-treat analysis, 407 in each arm. The ten-week mortality was 24.8% (101/407% CI 20.7–29.3%) in the AmBisome arm and 28.7% (117/407, 95% CI 24.4–33.4%) [21]. The absolute difference in 10-week mortality risk between the AmBisome arm and control arm was -3.9% and the upper limit of the one-sided 95% confidence interval for this mortality risk difference was 1.2%, indicating non-inferiority. When adjusting for factors independently associated with mortality the AmBisome regimen was found to be superior. In addition, the AmBisome regimen was associated with significantly fewer adverse events including anaemia requiring blood transfusion, thrombophlebitis and electrolyte abnormalities. Based on the trial findings, the World Health Organization updated their guidelines in early 2022 to recommend the single, high-dose of AmBisome given with 14 days of flucytosine and fluconazole as first-line therapy in resource limited settings [22].

Having proved the clinical efficacy, it is essential to consider the potential barriers and facilitators to real-world implementation of the AmBisome regimen. We conducted a qualitative methods study with the aim of understanding the acceptability of the AmBisome regimen compared with the standard of care from both the participant and the researcher perspective.

## Methods

### Ethics statement

This study was approved by the Human Resource Development Council, Gaborone (HPDME: 13/18/1); Makerere School of Health Sciences Institutional Review Board, Kampala (REF: 2019–061), Uganda National Council for Science and Technology (REF: SS386ES) and the London School of Hygiene and Tropical Medicine (REF: 17957). Written informed consent was obtained from all participants.

We embedded an ethnographic study entitled The Lived Experience Of Participants in an African RandomiseD trial (LEOPARD) within the AMBITION-cm trial at the Gaborone, Botswana and Kampala, Uganda sites [23]. Through LEOPARD we aimed to understand the experience of participating in the AMBITION-cm trial from a range of different perspectives. We conducted in-depth interviews (IDIs) and direct observations, collecting data from three categories of individuals: trial participants, surrogate decision makers (SDMs) who provided consent for the trial in cases where potential participants lacked decision making capacity, and researchers working on the trial. The qualitative methods study focused on several key aspects of the trial including decision-making around entry into the trial, the informed consent process, and the broader dynamics of the transnational research partnership within which the trial was conducted. In addition, we aimed to understand the acceptability of the intervention with a particular focus on participants, SDMs and those researchers who were directly providing clinical care.

Consecutively eligible trial participants were approached to participate in two in-depth interviews. In-depth interviews provide the opportunity for the conversation to flow, to ask follow-up questions, probe for additional information, and circle back to key questions later. In general, this approach provides richer, more in-depth data than structured interviews. We aimed to recruit a maximum of 20 participants from each site, 40 in total. We included individuals who upon entry into the trial were deemed to have decision making capacity (i.e. decision orientated) and those who were not (i.e. decision disorientated). We anticipated 30% of all trial participants to be disorientated at baseline but aimed for half of all participants in this qualitative study to have been disorientated. At the time of enrolment into LEOPARD all individuals must have regained decision making capacity. We aimed for roughly 50–60% of participants to be male, in line with the epidemiology of cryptococcal meningitis. The first IDI took place at least six weeks into the ten-week trial and the other at least four weeks after the final trial appointment. Secondly, consecutively eligible surrogate decision makers were approached to participate in a single in-depth interview at least six weeks after having provided consent for a trial participant. We aimed to recruit a maximum of 15 individuals from each site, 30 in total, with no specification for gender. Finally, we purposively selected a range of researchers working on the trial to participate in a single in-depth interview. We approached individuals with different roles including senior and junior researchers, research doctors and nurses, laboratory scientists, pharmacists and study coordinators. Our sample size was 12 for each site: 12 in Botswana, 12 in Uganda and 12 affiliated to collaborating European institutions, 36 in total.

Interviews followed a topic guide tailored to each group of participants (S1 File). The trial participant and surrogate decision maker topic guides explored the experience of developing cryptococcal meningitis (or caring for someone who had), being approached and deciding to enrol in the trial, and the experience whilst in the trial. The researcher topic guide focused on the day-to-day experience working on the trial and broader impressions of the AMBITION-cm trial and global health research in general. All interviews were audio-recorded. Additionally, we conducted direct observations of the research process, including the informed consent process and the administration of study drugs. Interviews were transcribed and translated, and field notes were made. These data were then entered into NVivo 12 and analysed using thematic analysis [24]. Thematic analysis involved six steps: familiarisation with data, initial code generation, searching for themes, reviewing themes, defining and naming themes and presenting final conclusions. Within this analysis we focus specifically on data from participants and those researchers who were providing direct care to trial participants as they had hands-on experience of providing the two different treatment regimens. When presenting data, the location, role, and gender of researcher participants is omitted because of the small number of eligible participants.

## Results

Between January 2020 and June 2021, we recruited a total of 89 individuals (Table 1)– 38 trial participants (18 in Gaborone, 20 in Kampala), 20 SDMs (9 in Gaborone, 11 in Kampala) and 31 researchers (11 in Gaborone, 9 in Kampala and 11 from European collaborating

**Table 1. Summary of trial participants and surrogate decision makers (SDMs) recruited into the qualitative methods study.**

|  | Age | Gender | Nationality | Language of interview | Education Level | Decision-making capacity | Trial Arm | Number of Interviews | SDM Interview | SDM Gender |
|---|---|---|---|---|---|---|---|---|---|---|
| Gaborone | 34 | Male | Batswana | Setswana | Secondary | Disorientated | AmBisome | 2 | Yes | Female |
|  | 50 | Male | Batswana | Setswana | Primary | Disorientated | AmBisome | 2 | Yes | Female |
|  | 44 | Male | Batswana | Setswana | Secondary | Disorientated | Control | 2 | Yes | Female |
|  | 34 | Female | Batswana | Setswana | Secondary | Disorientated | Control | 1 | Yes | Female |
|  | 32 | Female | Batswana | Setswana | Tertiary | Disorientated | Control | 2 | Yes | Female |
|  | 49 | Male | Batswana | Setswana | Tertiary | Disorientated | Control | 2 | Yes | Female |
|  | 35 | Male | Zimbabwean | English | Secondary | Disorientated | AmBisome | 2 | Yes | Female |
|  | 44 | Female | Batswana | Setswana | Tertiary | Disorientated | AmBisome | 1 | No |  |
|  | 34 | Male | Zimbabwean | English | Secondary | Disorientated | AmBisome | 1 | No |  |
|  | 37 | Female | Zimbabwean | English | Secondary | Orientated | Control | 1 |  |  |
|  | 24 | Female | Zimbabwean | English | Secondary | Orientated | Control | 2 |  |  |
|  | 42 | Male | Batswana | Setswana | Secondary | Orientated | AmBisome | 2 |  |  |
|  | 37 | Male | Batswana | Setswana | Secondary | Orientated | AmBisome | 2 |  |  |
|  | 40 | Male | Batswana | Setswana | Secondary | Orientated | AmBisome | 2 |  |  |
|  | 47 | Male | Zimbabwean | English | Secondary | Orientated | AmBisome | 2 |  |  |
|  | 22 | Male | Batswana | Setswana | Secondary | Orientated | Control | 2 |  |  |
|  | 33 | Female | Batswana | Setswana | Secondary | Orientated | AmBisome | 2 |  |  |
|  | 29 | Female | Zimbabwean | English | Primary | Orientated | Control | 1 |  |  |
| Kampala | 46 | Female | Ugandan | Luganda | Primary | Disorientated | Control | 2 | Yes | Female |
|  | 53 | Female | Ugandan | Luganda | Primary | Disorientated | AmBisome | 2 | Yes | Female |
|  | 26 | Female | Ugandan | Luganda | Primary | Disorientated | Control | 1 | Yes | Male |
|  | 29 | Female | Ugandan | Luganda | Secondary | Disorientated | AmBisome | 1 | Yes | Female |
|  | 36 | Male | Ugandan | Luganda | Primary | Disorientated | Control | 2 | Yes | Female |
|  | 35 | Male | Ugandan | Luganda | Primary | Disorientated | Control | 2 | Yes | Female |
|  | 45 | Male | Ugandan | Luganda | Tertiary | Disorientated | Control | 2 | Yes | Female |
|  | 35 | Male | Ugandan | Luganda | Primary | Disorientated | AmBisome | 2 | Yes | Female |
|  | 30 | Female | Ugandan | Luganda | Secondary | Disorientated | Control | 2 | Yes | Female |
|  | 27 | Male | Ugandan | Luganda | Primary | Disorientated | AmBisome | 2 | Yes | Male |
|  | 49 | Male | Ugandan | Luganda | Primary | Orientated | AmBisome | 2 |  |  |
|  | 44 | Male | Ugandan | Luganda | Primary | Orientated | Control | 1 |  |  |
|  | 24 | Male | Ugandan | Luganda | Secondary | Orientated | AmBisome | 2 |  |  |
|  | 46 | Female | Ugandan | Luganda | Primary | Orientated | Control | 2 |  |  |
|  | 45 | Male | Ugandan | Luganda | Primary | Orientated | Control | 2 |  |  |
|  | 32 | Female | Ugandan | Luganda | Secondary | Orientated | AmBisome | 2 |  |  |
|  | 34 | Female | Ugandan | Luganda | Tertiary | Orientated | AmBisome | 2 |  |  |
|  | 23 | Female | Ugandan | Luganda | Primary | Orientated | Control | 2 |  |  |
|  | 23 | Female | Ugandan | Luganda | Primary | Orientated | AmBisome | 2 |  |  |
|  | 30 | Male | Ugandan | Luganda | Secondary | Orientated | Control | 2 |  |  |

Three male surrogate decision makers were interviewed without being linked to a trial participant due to the ill health of the trial participant: two in Gaborone and one in Kampala.

institutions). Forty-eight (54%) of the participants were female. Initial interviews ranged in duration from 20 to 163 minutes with a median duration of 52 minutes.

## The perspective of participants

Most participants had long, convoluted pathways through care leading to their diagnosis of cryptococcal meningitis. The vast majority had a headache at the time of diagnosis, and they had often navigated through multiple healthcare facilities prior to reaching the AMBITION-cm site hospital. During this time, they had experienced a gradual deterioration in health and mental status, common to cryptococcal meningitis, such as the development of disturbing hallucinations, seizures, confusion and reduced consciousness. As a result, we found that the decision to enrol in the trial was predominantly motivated by fear of death, an acknowledgment that the trial was their best chance of survival, and trust in the research teams. This subject has been discussed in more detail elsewhere [25]. The levels of comprehension around the trial aims and design were relatively low and participants found it difficult to disentangle the different parts of their treatment. For example, when asked if they knew that some *'were given one yellow bottle [of amphotericin] while others were given seven'*, one 37 year old male participant in Gaborone responded saying *'I did not know about that'* whilst a 48 year old female participant in Kampala explained they *'did not know because there was a time when I had lost my senses'* and a 35 year old male participant in Kampala said that *'the truth is that I may have been unconscious'*. This resulted in a limited amount of primary data around the acceptability of the AmBisome regimen directly from trial participants and instead we had to rely more on the testimonies of the researchers who also made this observation:

> *Interviewer: Do you think that the patients appreciate that there is a difference between the treatments that are on offer? That there's the control arm and then there's the single dose arm?*
>
> *Researcher: I doubt they appreciate, they notice that, I doubt. I think they notice more the interactions than the actual medicine.*

We found a general preference among participants for the AmBisome dose arm, due to an aversion to having multiple intravenous doses as described by a doctor who said *'they think they would have the drip in only for one day . . . so they'd rather have the single dose over the seven'*. When considering if there were any concerns about only getting one dose versus the seven offered in the control arm, albeit of a different formulation, there was one mention from a doctor of potential concerns as this arm was sometimes referenced as being the *'experimental arm'* within the trial and therefore carrying an element of uncertainty. Nevertheless, they felt that most participants had confidence in the single dose, and this was confirmed by several participants in Kampala who were managed on an open ward and able to see the progress of others in the trial. For example, one 32 year old female participant felt that *'the one bottle works quickly because I realised that the others who were given the seven bottles could take some time to respond to the treatment'* and a 23 year old female participant had a similar observation:

> *'I noticed that others who were getting seven bottles lacked strength and were very weak and were being supported when walking and I, who had received one bottle, was stronger than my colleagues. So, I did not long for the seven bottles because my health condition improved quickly but those patients who received many bottles still had a weak health condition.'*

Although the trial itself did not find a difference in the duration of hospitalisation between arms, participants were highly motivated by fewer intravenous infusions and the potential prospect of shorter admissions and therefore being able to get back to work and/or other household responsibilities. One doctor told us that trial participants *'don't doubt the one dose'* and thought that having the single dose meant that `*they would get to leave hospital maybe day eight. Because the ones who get a single dose sometimes think they will go home soon after they get the AmBisome.'*

### Researcher observations on administration of the two treatment regimens

There were significant differences between the two regimens in terms of the medication given and the time required to do this (Fig 1). Researcher participants found that the single, high-dose of AmBisome took longer to reconstitute. On average 12 vials were used per participant and reconstitution was reported to take between 20 and 40 minutes. The infusion ran over two hours and had to be preceded and followed by a litre of intravenous normal saline. Participants received twice daily doses of oral potassium supplementation and a daily dose of magnesium supplementation for each day they received amphotericin and two additional days, so three days in the single-dose arm. In addition, fluconazole was given daily and flucytosine 6 hourly for 14 days.

In comparison, in the control arm the conventional amphotericin B deoxycholate took roughly 5 to 10 minutes to reconstitute and was given over a four-hour infusion. The pre- and post-hydration was required for the seven days of amphotericin therapy and the oral electrolyte supplementation for nine. With regards to the oral antifungals, the 6 hourly flucytosine was only given for seven days and was followed by seven days of fluconazole.

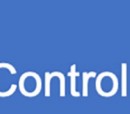

**Single dose**
- 20-40 minutes to reconstitute
- 2 hour infusion
- Pre- and post- hydration for 1 day
- Electrolyte supplementation for 3 days
- Fluconazole daily for 14 days AND
- Flucytosine four times daily for 14 days

**Control**
- 5-10 minutes to reconstitute
- 4 hour infusion
- Pre- and post- hydration for 7 days
- Electrolyte supplementation for 9 days
- Flucytosine four times daily for 7 days THEN
- Fluconazole daily for 7 days

**Fig 1. A summary of how each of the two treatment regimens were administered.**

### Researcher perspectives on managing the two treatment regimens

Drawing on the experience of the research teams looking after participants there was a clear preference for the AmBisome arm (Fig 2). It was felt to be worth the time investment of the initial efforts to prepare the large number of vials. A nurse told us:

> *'I like the single dose because it's less work . . . it means you give them medication one day and unlike putting ampho[tericin B] every day for 7 days, and ampho[tericin] also has its own dynamics, you need to pre-hydrate every day . . . sometimes you come to the hospital and you find that the patient is not pre-hydrated, now you start pre-hydrating first, sometimes you come to the hospital, the cannula has tissued, you need to start to putting in cannulae, you know all those dynamics of giving ampho[tericin] on the daily.'*

While individual doses of conventional amphotericin were easier to reconstitute than AmBisome doses, the seven-day course and the issues with fluids and intravenous lines were felt to overshadow this. One drawback of the single-dose arm was that the 6 hourly dosing of flucytosine (given for 14 days in the single-dose arm compared to 7 days in the control arm) was found to be inconvenient for participants who had to set alarms in the night or remind one another to take their dose, and so the shorter duration of flucytosine in the control arm was a positive as described here by a nurse:

> *'I wouldn't have liked it (the flucytosine), especially that 4am dose, but these patients came to cope with it because we had explained to them how complicated the disease is, how missing doses would cause them problems and stuff like that, how it was dangerous to miss doses, then the side effect profile, what would happen. So many of them actually welcomed the idea of taking doses as prescribed and they actually got to figure out how to liaise within themselves.*

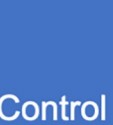

**Single dose**
- Clear preference for the single-dose arm
- 'Worth the investment'
- Fewer days of additional hydration
- Fewer tissued cannulae
- Rigors less common
- Less drug-induced toxicity
- Fewer monitoring bloods as a result

**Control**
- Quicker to reconstitute but overall a lot more work
- Only seven days of flucytosine which is difficult, particularly at night
- Thrombophlebitis more common
- Lots more cannulae required
- More time consuming
- Patients become aware of the drug toxicity
- Some asking why they did not get the single dose
- Difficult to access blood transfusions

**Fig 2. Feedback on the administration and toxicity of the two treatment regimens.**

*They made kind of a system within themselves that was motivated by the study nurses that they would remind each other'*

With regards to toxicity and management, the researchers consistently stated that they observed fewer cases of amphotericin induced rigors in the single-dose arm. As was later proven with the formal trial analysis they also observed less drug-induced toxicity and were pleased to have less work managing individuals who developed toxicities. In general, participants treated with the control arm were found to be more time consuming. Researchers also found that it was very difficult to access blood transfusions which were required more frequently in the control arm.

*'Less admin, less side-effects, because the patients would be having rigors, then I will have to deal with it (laughs) but if it's, you know, I have to deal with the toxicities, write lots of adverse events, so, it's less work for me if it's AmBisome, it's nicer for the patient also. I don't have to be changing cannulas on the patient every other day so it's really nice for everyone. The nurses don't have to stay here long, waiting for the 4 hours of amphotericin. We love AmBisome.'*

When asked if the participants noticed any difference between the arms in terms of toxicity, researchers in Kampala said that some participants in the control arm became aware of the toxicity they experienced and attributed this to the yellow amphotericin, as described by this research doctor.

*'Of course, most of them if they get the control they would be like, "Oh I wish I had gotten a single dose", especially if they get phlebitis like on day three and they start saying, "Oh I wish I had gotten one dose of this yellow medicine" . . . because they notice that people who get a single dose, their arms are never swollen.'*

These findings were consistent with the primary trial analysis which found that thrombophlebitis requiring antibiotic therapy was more common in the control arm and from the perspective of researchers this added to the list of recurring problems with intravenous lines.

## Discussion

We found that the single, high-dose AmBisome regimen was acceptable to both participants and researchers within the AMBITION-cm trial. The AmBisome regimen was more time consuming to prepare on day one, but this was felt to be a worthwhile investment because of the additional time required to administer the additional doses of amphotericin B deoxycholate and to avert and manage amphotericin-related toxicity. Participants in the control arm were observed to suffer more regularly from amphotericin induced rigors and thrombophlebitis which often required a lot of medical input, particularly in terms of intravenous access. In addition to the health impact on participants, the increased drug-related toxicity observed in the control arm was time consuming for researchers to manage, required additional resources, and was sometimes difficult to resolve, particularly in terms of the limited availability of blood for transfusions.

The AMBITION-cm trial was well staffed and resourced with external funding. In routine care settings with a high incidence of HIV-associated cryptococcal meningitis healthcare workers are often caring for large numbers of patients with a range of complex medical conditions. In addition, healthcare facilities may not always have access to the resources required to both avert and manage drug-related toxicities. As a result, we believe it is reasonable to assume that the challenges encountered by our research team when managing participants in the

control arm would be amplified in routine care settings. There is extensive evidence demonstrating that outcomes of individuals diagnosed with cryptococcal meningitis are worse in routine-care settings compared to within clinical trials, even when receiving the same antifungal treatment regimen [26]. Although the reasons behind this are multi-factorial, the time, expertise and resources required to avert and manage amphotericin-related toxicity is a key driver of this difference. One key rationale behind the AMBITION-cm trial regimen was to identify an effective but also safe and easier to administer treatment for cryptococcal meningitis. This study complements the main trial efficacy data in that respect and demonstrates that the single, high-dose AmBisome regimen was much simpler to administer and manage. When considering widespread implementation within stretched healthcare systems the true benefits of the AmBisome regimen are therefore also likely to be amplified.

There are limitations to this study. Participants were purposively recruited following a sampling matrix based on gender and severity of infection at baseline but these results are not intended to be fully representative or generalisable. The participants and SDMs themselves had their own unique experience being treated with one arm so it was clearly not possible to fully explore their preferences for one over another. In addition, due to the severity of their unfolding, life-threatening illness participants found it difficult to disentangle the different parts of their treatment which made it challenging to elicit their perspectives on the different arms. We therefore relied heavily on the data collected from researchers. Finally, we acknowledge the positionality of the research team, including DSL as the Chief Investigator of this study and the lead clinician of the AMBITION-cm trial, and how this may have resulted in some desirability bias and a Hawthorne effect during data collection. We aimed to overcome this by forming a research group including social scientists external to the trial who collected the data from participants and SDMs in Gaborone and Kampala.

In conclusion we found that the single, high-dose AmBisome regimen was highly acceptable to both participants and researchers in the clinical trial. These findings complement the clinical efficacy data from the clinical trial to support widespread implementation of the regimen.

## Supporting information

**S1 File. In-depth interview schedules.**
(PDF)

## Acknowledgments

We are greatly indebted to the individuals who participated in this study, particularly those who were recovering from a severe illness. We thank all of the research and routine care staff at each site who helped care for the participants and co-facilitated this research, particularly during the COVID-19 pandemic. Thanks to expert patients and Community Advisory Board members across the sites for their important feedback on the design and methodology of this study. Finally, we acknowledge Dr Agatha Bula, Dr Graeme Hoddinott, Dr Zivai Mupambireyi and Dr Deborah Nyirenda for their early comments and input to the study design and data collection tools.

## Author Contributions

**Conceptualization:** David S. Lawrence, Agnes Ssali, Janet Seeley.

**Data curation:** David S. Lawrence, Agnes Ssali, Neo Moshashane, Georgina Nabaggala.

**Formal analysis:** David S. Lawrence, Agnes Ssali.

**Funding acquisition:** Joseph N. Jarvis.

**Investigation:** David S. Lawrence, Agnes Ssali, Neo Moshashane, Georgina Nabaggala, Joseph N. Jarvis.

**Methodology:** David S. Lawrence, Agnes Ssali, Thomas S. Harrison, David B. Meya, Joseph N. Jarvis, Janet Seeley.

**Project administration:** David S. Lawrence, Agnes Ssali, Neo Moshashane, Georgina Nabaggala, Lebogang Maphane, David B. Meya.

**Resources:** David S. Lawrence.

**Software:** David S. Lawrence.

**Supervision:** David S. Lawrence, Thomas S. Harrison, David B. Meya, Joseph N. Jarvis, Janet Seeley.

**Validation:** David S. Lawrence.

**Visualization:** David S. Lawrence.

**Writing – original draft:** David S. Lawrence.

**Writing – review & editing:** Agnes Ssali, Neo Moshashane, Georgina Nabaggala, Lebogang Maphane, Thomas S. Harrison, David B. Meya, Joseph N. Jarvis, Janet Seeley.

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
