## [Decision Letter · Decision Letter 0]

2 Sep 2022

Dear Dr. Lawrence,

Thank you very much for submitting your manuscript "The acceptability of the AMBITION-cm treatment regimen for HIV-associated cryptococcal meningitis: findings from a qualitative methods study of participants and researchers in Botswana and Uganda" for consideration at PLOS Neglected Tropical Diseases. As with all papers reviewed by the journal, your manuscript was reviewed by members of the editorial board and by two independent reviewers with relevant expertise. The reviewers appreciated the attention to an important topic. Based on the reviews, we are likely to accept this manuscript for publication, providing that you modify the manuscript according to the review recommendations. 

Sincerely,

Chaoyang Xue, Ph.D.

Academic Editor

Elsio Wunder Jr

Section Editor

Reviewer's Responses to Questions

**Key Review Criteria Required for Acceptance?**

**Methods**

-Are the objectives of the study clearly articulated with a clear testable hypothesis stated?

-Is the study design appropriate to address the stated objectives?

-Is the population clearly described and appropriate for the hypothesis being tested?

-Is the sample size sufficient to ensure adequate power to address the hypothesis being tested?

-Were correct statistical analysis used to support conclusions?

-Are there concerns about ethical or regulatory requirements being met?

Reviewer #1: (1) consider including the interview questions into supplemental materials (were the same interview questions used in all study sites?) 

(2) Please describe steps taken to ensure that the interviewer was not inadvertently influencing the participant’s opinion in favor of one arm versus the other.

Reviewer #2: The method is in order

**Results**

-Does the analysis presented match the analysis plan?

-Are the results clearly and completely presented?

-Are the figures (Tables, Images) of sufficient quality for clarity?

Reviewer #1: (3) If possible, consider showing the participants’ demographic information at each study site, including the language. These data can be organized in a table or graph format 

(4) Instead of using subjective terms like “general preference”, “most participants”, “clear preference”, etc., please consider including the actual numbers of participants favoring one arm of the trial over the other… Is it possible to assign numbers to these terms?

Reviewer #2: The results are in order

**Conclusions**

-Are the conclusions supported by the data presented?

-Are the limitations of analysis clearly described?

-Do the authors discuss how these data can be helpful to advance our understanding of the topic under study?

-Is public health relevance addressed?

Reviewer #1: (No Response)

Reviewer #2: The conclusiond drawn form the study based on the set out objectives were in order.

The research limitations were also discussed.

**Editorial and Data Presentation Modifications?**

Reviewer #1: (No Response)

Reviewer #2: The conclusion from the paper is in line with known knowledge evidenced by the WHO 2022 guideline and the paper (ref 21) the authors cited.

The extra information the authors provided was evidence for acceptability of Ambisome regimen I am not sure if this is enough data for a full research article.

The sample size is very small

**Summary and General Comments**

Reviewer #1: The manuscript by Lawrence et al describes results of a qualitative follow up study to the Ambisome Therapy Induction Optimization clinical trail aimed at investigating the acceptance of the treatment regiments by patients, their caregivers and healthcare workers participating in the clinical trial. The study is based on the interviews and direct observations. The manuscript is well written, the results are important. In my opinion, the paper will benefit from providing more concrete data in support the observed tendencies. A few suggestions are below:

(1) consider including the interview questions into supplemental materials (were the same interview questions used in all study sites?) 

(2) Please describe steps taken to ensure that the interviewer was not inadvertently influencing the participant’s opinion in favor of one arm versus the other. 

(3) If possible, consider showing the participants’ demographic information at each study site, including the language, in which the interview was conducted. These data can be organized in a table or graph format 

(4) Instead of using subjective terms like “general preference”, “most participants”, “clear preference”, etc., please consider including the actual numbers of participants favoring one arm of the trial over the other… Is it possible to assign numbers to these terms?

Reviewer #2: The paper is good and well-written.

PLOS authors have the option to publish the peer review history of their article (what does this mean?). If published, this will include your full peer review and any attached files.

Reviewer #1: No

Reviewer #2: No

Figure Files:

Data Requirements:

Reproducibility:

References

---

## [Editor Report · Decision Letter 1]

16 Sep 2022

Dear Dr. Lawrence,

We are pleased to inform you that your manuscript 'The acceptability of the AMBITION-cm treatment regimen for HIV-associated cryptococcal meningitis: findings from a qualitative methods study of participants and researchers in Botswana and Uganda' has been provisionally accepted for publication in PLOS Neglected Tropical Diseases.

Best regards,

Chaoyang Xue, Ph.D.

Academic Editor

Elsio Wunder Jr

Section Editor

---

## [Editor Report · Acceptance letter]

19 Oct 2022

Dear Dr. Lawrence,

We are delighted to inform you that your manuscript, "The acceptability of the AMBITION-cm treatment regimen for HIV-associated cryptococcal meningitis: findings from a qualitative methods study of participants and researchers in Botswana and Uganda," has been formally accepted for publication in PLOS Neglected Tropical Diseases.

Best regards,

Shaden Kamhawi

co-Editor-in-Chief

Paul Brindley

co-Editor-in-Chief
